# Plants and Small Molecules: An Up-and-Coming Synergy

**DOI:** 10.3390/plants12081729

**Published:** 2023-04-21

**Authors:** A. Lepri, C. Longo, A. Messore, H. Kazmi, V. N. Madia, R. Di Santo, R. Costi, P. Vittorioso

**Affiliations:** 1Department of Biology and Biotechnology “Charles Darwin”, Sapienza University of Rome, 00185 Rome, Italy; andrea.lepri@uniroma1.it (A.L.); chiara.longo@uniroma1.it (C.L.); hira.kazmi@uniroma1.it (H.K.); 2Department of Chemistry and Technology of Drug, Istituto Pasteur Italia—Fondazione Cenci Bolognetti, Sapienza University of Rome, Piazzale Aldo Moro 5, 00185 Rome, Italy; antonella.messore@uniroma1.it (A.M.); valentinanoemi.madia@uniroma1.it (V.N.M.); roberto.disanto@uniroma1.it (R.D.S.); roberta.costi@uniroma1.it (R.C.)

**Keywords:** chemical genetics, small molecules, abiotic stress, epigenetics, development

## Abstract

The emergence of Arabidopsis thaliana as a model system has led to a rapid and wide improvement in molecular genetics techniques for studying gene function and regulation. However, there are still several drawbacks that cannot be easily solved with molecular genetic approaches, such as the study of unfriendly species, which are of increasing agronomic interest but are not easily transformed, thus are not prone to many molecular techniques. Chemical genetics represents a methodology able to fill this gap. Chemical genetics lies between chemistry and biology and relies on small molecules to phenocopy genetic mutations addressing specific targets. Advances in recent decades have greatly improved both target specificity and activity, expanding the application of this approach to any biological process. As for classical genetics, chemical genetics also proceeds with a forward or reverse approach depending on the nature of the study. In this review, we addressed this topic in the study of plant photomorphogenesis, stress responses and epigenetic processes. We have dealt with some cases of repurposing compounds whose activity has been previously proven in human cells and, conversely, studies where plants have been a tool for the characterization of small molecules. In addition, we delved into the chemical synthesis and improvement of some of the compounds described.

## 1. Introduction

From the classical genetic studies on peas by Gregor Mendel to the present, molecular strategies and technologies in plant biology have undergone remarkable progress, from gene knockdown by antisense RNA to RNA interference and from gene knockout by T-DNA or transposons to genome editing. This has led to a significant growth in the knowledge of the molecular mechanisms underlying plant growth and development as well as plant response to biotic and abiotic stress, both in the model organism *Arabidopsis thaliana* and in plants of agronomic interest such as tomato, soybean, and rice. Nevertheless, studies on the function of essential genes have been hampered by a lack of viable mutants due to lethality of gene function or by mutants with a pleiotropic phenotype due to functional redundancy. Despite many different strategies that have been developed to overcome the major drawbacks of conventional genetic approaches, these still represent an important limit in terms of time and efficiency. One branch able to fill this gap is chemical genetics, which relies on small molecules to phenocopy genetic mutations addressing specific targets. Chemical genetics allows us to study the function of a gene product in its cellular/organism context using exogenous ligands. In this approach, small molecules, ligands, or drugs directly bind to protein targets, thus altering protein function and mimicking the phenotype of mutant(s) of the corresponding gene(s). Their application at increasing concentrations can result in a wider spectrum of phenotypes compared to the study of knockout mutants, while removal of the compound or addition of an antagonist will easily revert the phenotype. Furthermore, the use of sub-lethal concentrations will allow the study of essential proteins whose knockout mutants are lethal. Since proteins can vary in terms of their primary, secondary, and tertiary structure as well as their post-translational modifications, the potentiality of binding sites for small molecules is extremely wide. In addition, given that Arabidopsis is also a worthy model in chemical genetics, it should be underlined that studies on small molecules effective on Arabidopsis open new, interesting perspectives for their application on genetically unfriendly species and for studying homologous genes [1,2].

Advances in recent decades have greatly enhanced both target specificity and activity, broadening the application of this approach to any biological process. As for classical genetics, chemical genetics also proceeds with a forward or reverse approach, depending on the nature of the study [3]. Forward chemical genetics is based on the screening of a huge number of ligands in order to study one phenotype of interest to identify the gene product(s) responsible for the phenotype following binding to the small molecule. In the reverse approach, the activity of a specific protein is blocked, reduced, or impaired by binding to a ligand, and any resulting phenotype can be investigated [4,5] (Figure 1).

In this review, we use the topic of chemical genetics as a tool for studying plant photomorphogenesis, stress responses and epigenetic processes (Figure 2). In addition, we deem the plant system as a tool for the identification of small molecules, and we deepen the chemical synthesis and improvement of some of the described compounds. According to this scheme, in this review we will discuss some of the relevant papers published in the last decade.

**Figure 1 plants-12-01729-f001:**
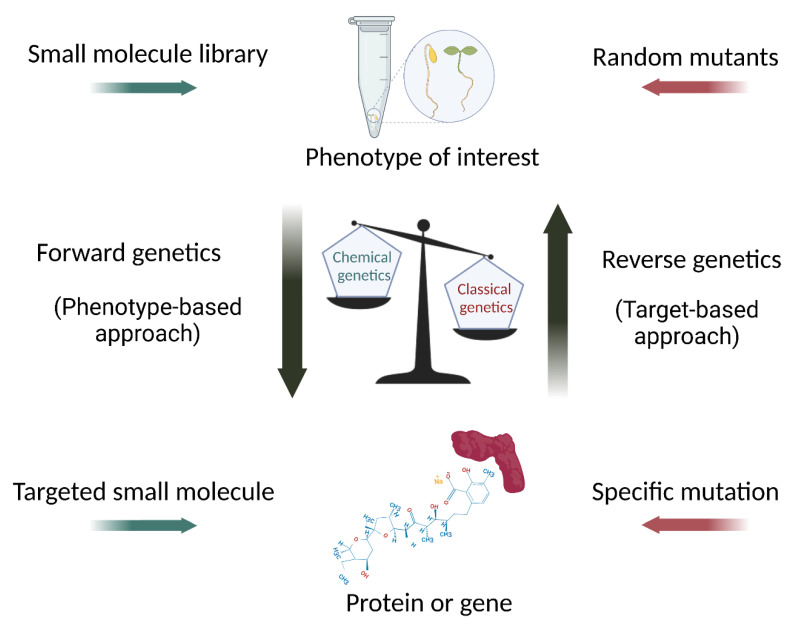
Comparison between the chemical and the classical genetic strategies. For both forward and reverse approaches, chemical genetics can be considered as a powerful strategy complementary to classical genetics, providing a useful tool to avoid related weaknesses. Nonetheless, classical genetics still has more weight than chemical genetics, for both technical and historical reasons. Created with BioRender.com (accessed on 6 April 2023).

## 2. A Successful Case: Small Molecules to Study Auxin

One of the most successful stories in plant chemical genetics concerns the study of the phytohormone auxin (indole-3-acetic acid; IAA) and of its receptors, signalling and trafficking molecules [6,7]. Indeed, auxin acts through a complex combinatorial signalling and trafficking system, which allows, through an auxin gradient, specific single cell responses to IAA at both a transcriptional and physiological level. The three main elements of the auxin signalling mechanism are the F-box proteins TIR1/AFB (TRANSPORT INHIBITOR RESISTANT1/AUXIN SIGNALING F-BOX), Aux/IAA (AUXIN/INDOLE-3-ACETIC ACID) acting as corepressors, and the transcription factors ARF (AUXIN RESPONSE FACTOR).

The first molecule issued by a chemical screening for auxin mutants was the herbicide 2,4-D (2,4-dichlorophenoxyacetic acid), a more stable analogue of IAA [8].

Two other compounds, NAA (naphthalene-1-acetic acid) and picloram, previously used in agriculture, have been employed as auxin-like synthetic substances. Subsequently, two independent chemical screenings identified two distinct molecules, characterized by common structural features acting as “proauxins” [9,10]. The crystallographic study revealing the structure of the TIR1-IAA complex suggested a certain degree of promiscuity of the auxin binding pocket, thus making it able to recognize different synthetic molecules [11,12]. Auxin functions in the control of organ development during embryogenesis as well as in settling the ratio between cell division and cell differentiation, mainly through the formation of local auxin gradients. A sophisticated system of influx and efflux auxin carriers ensures the local distribution of the hormone through the polar localization of these transmembrane molecules [13,14]. This is based on a tightly controlled endomembrane trafficking system, which was also unravelled thanks to chemical genetics. The natural compound BFA (brefeldin A), a fungal toxin, is, among many chemical compounds, the one most widely used to explore endocytosis and the recycling of auxin carriers. BFA inhibits the key regulators of vesicle formation both in animals and plants, i.e., the ARF-GEFs (ADP ribosylation factor guanine nucleotide exchange factors) [15,16]. The mechanism of action of the Arabidopsis ARF-GEF protein GNOM in the trafficking and recycling of the PIN1 (PIN-FORMED 1) auxin efflux carrier has been indeed unveiled by using BFA as a vesicle trafficking inhibitor joined with the auxin polar transport inhibitor TIBA (2,3,5-triiodobenzoic acid) [17,18]. Since the first studies with BFA, the use of chemical molecules has been instrumental in unravelling the endomembrane trafficking system in plants. Both reverse chemical genetics and forward approaches have been used to identify inhibitors of membrane endocytosis [19] and vacuolar trafficking [20,21]. Since this is a very broad field of research, it will not be detailed in this review, but we refer to exhaustive reviews on this topic [5,22,23,24,25].

**Figure 2 plants-12-01729-f002:**
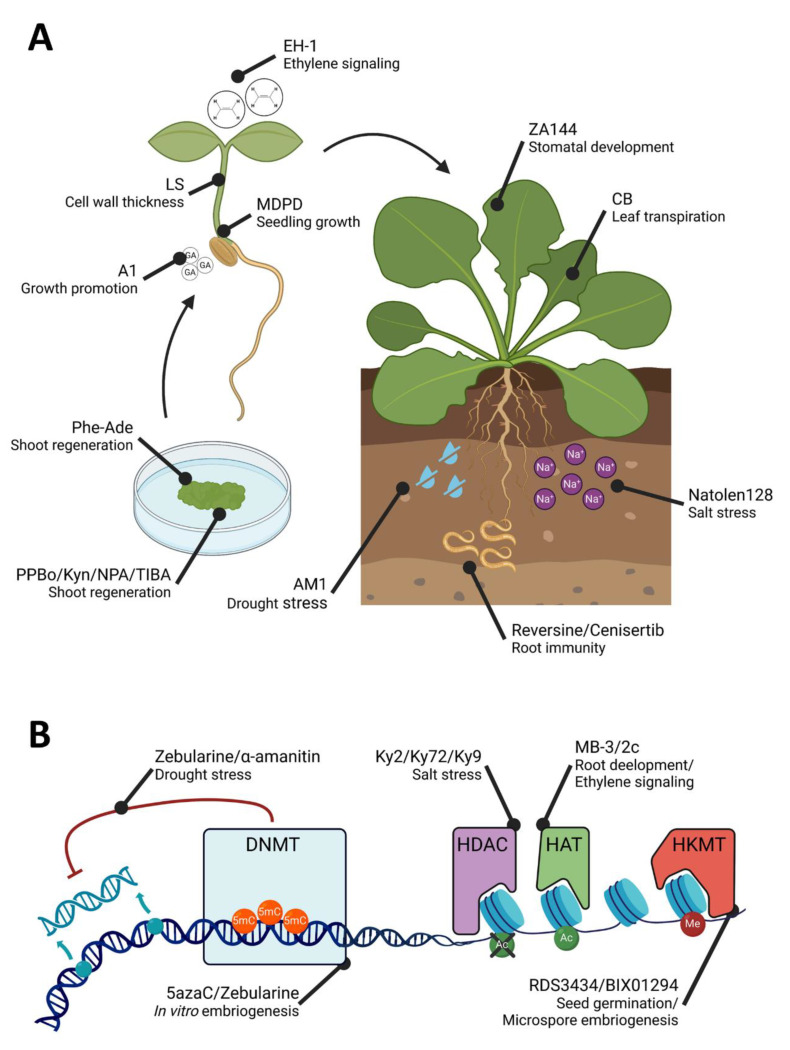
Small molecules involved in plant morphogenesis and stress response (**A**) and epigenetic processes (**B**). The figure shows different molecules described in the review and the processes in which they are involved. Created with BioRender.com (accessed on 6 April 2023).

## 3. Small Molecules to Gain Insights into Plant Morphogenesis and Development

Photomorphogenesis occurs once germination is completed, in the presence of light. This complex developmental program involves an extensive remodelling of seedling morphology aimed at optimizing light capture to activate photosynthesis and in turn enabling autotrophy. Indeed, photomorphogenic growth is characterised by the inhibition of hypocotyl elongation, the opening of the apical hook, cotyledon expansion, chloroplast differentiation, and the activation of shoot and root apical meristems. Photomorphogenic seedlings display a short and green shoot with expanded leaves, and a well-developed root system and seedling organs, namely root, hypocotyl and cotyledons, have been widely exploited for molecular genetic studies [26].

In Arabidopsis, NAEs (*N*-acylethanolamines), a group of fatty acid amides conserved among eukaryotes [27], play a role in early seedling development [28]. Indeed, it has been shown that the addition of NAE 12:0 (*N*-lauroylethanolamine) causes an inhibition of seedlings [28]; NAE 12:0 is degraded into ethanolamine and free fatty acids by FAAH (FATTY ACID AMIDE HYDROLASE) [29]. Indeed, it has been observed that a lack of FAAH from Arabidopsis (AtFAAH) results in a more severe growth inhibition by NAE 12:0 in *Atfaah*-knockout seedlings, whereas *AtFAAH*-overexpressing seedlings were more resistant compared to the wild type [30]. A recent study has developed a strategy using small molecules to counteract the effects of NAEs. In vertebrates, NAEs contribute to the endocannabinoid signalling pathway, and therefore, they influence physiological states such as appetite, mood, and sleep, as well as cardiovascular function and reproduction [31]. Given that NAEs are well conserved between animals and plants, and given the availability of a repertoire of active molecules against NAEs, Khan and collaborators screened a library of 10,000 synthetic compounds with the aim of identifying small molecules able to reduce the growth-inhibitory effect of NAE12:0 on Arabidopsis seedling [32]. Based on the efficacy to rescue NAE12:0-induced phenotypes of root length, root hairs and cotyledon area, only one compound was identified—the 6-(2-methoxyphenyl)-1,3-dimethyl-5-phenyl-1*H*-pyrrolo[3,4-*d*]pyrimidine-2,4(3 *H*,6 *H*)-dione, named MDPD (Figure 3). Whereas the rescue was only partial for primary root length, it was complete both for root hairs and the cotyledon expansion phenotype [32]. In vitro amidohydrolase activity on the recombinant AtFAAH and FAAH from rat (ratFAAH) clearly revealed that MDPD is able to increase the enzymatic efficiency of AtFAAH (3-to-4 fold) but not of ratFAAH, suggesting that MDPD is specifically active on plant FAAH. [32]. Nevertheless, since MDPD can partially rescue the hypersensitive phenotype NAE12:0 of the *Atfaah* mutant seedlings, it seems likely that, in the absence of AtFAAH, it may function also by enhancing the activity of other amidohydrolases.

The phytohormones known as GAs (gibberellins) are involved in plant growth as growth-promoting hormones and in plant development, as GAs strictly control the seed-to-seedling as well as the vegetative-to-reproductive transition phases [33,34]. In particular, this family of tetracyclic diterpenoid compounds promotes organ growth through the enhancement of cell division and cell expansion [35]. GAs’ perception is mediated by the GA receptor GID1 (GIBBERELLIN INSENSITIVE DWARF1), while key components of the GAs’ signalling pathway include the DELLA (aspartic acid–glutamic acid–leucine–leucine–alanine) growth inhibitors (DELLAs) and the F-box proteins SLY1 (SLEEPY1) and SNZ (SNEEZY) in Arabidopsis [33,36]. Degradation of DELLA proteins, following their binding to the GA-GID1 complex, causes the inhibition of GA-mediated processes [37]. The research interest on GAs has been continuously increasing since the so-called “green revolution” due to their growth-promoting action, which is particularly relevant in agriculture to enhance crop yields. By means of a chemical approach, a naphthalene sulfonamide compound named A1 (Figure 3) has recently been identified for its growth-promoting activity [38]. This small molecule is a non-chlorinated analogue of the naphthalene sulfonamide compounds W5 (*N*-(6-aminohexyl)-1-naphthalenesulfonamide) and W7 (*N*-(6-aminohexyl)-5-chloro-1-naphthalenesulfonamide), previously identified as calmodulin inhibitors [39] and used in plants as growth-promoting chemicals. Although W7 differs only in the chlorine C-5 substituent in respect to W5, thus displaying a high degree of structural similarity, these compounds showed different efficacy in terms of Ca^2+^ response, with W5 being significantly less effective than W7 [39]. Therefore, more recently, different analogues of W5 and W7 were synthesized and tested in plants [38]. In contrast to all the new calmodulin inhibitors, the A1 small molecule was able to promote root growth instead of inhibiting it. Since this effect was not root-specific, it was investigated whether the A1 molecule could act through the GA/DELLA signalling pathway. Treatment with A1, similarly to bioactive GA, resulted in the degradation of the RGA-GFP (REPESSOR OF *ga1*-GFP) chimeric protein, thus suggesting the hypothesis that A1 may act as a GA analogue. To further prove that A1 functions through the GA-DELLA molecular mechanism, the authors assessed the effect of this compound on the penta *della* mutant, which lacks the five Arabidopsis DELLA proteins and is characterised by a longer hypocotyl [40]. While treatment with A1 promoted the hypocotyl growth of wild type seedlings, similarly to GAs, it had no effect on *della* mutant seedlings due to the lack of the DELLA proteins, thus confirming that A1 requires DELLA proteins for its growth-promoting effect. A similar experiment with the GA biosynthetic mutant *ga1-5*, which has low levels of GA and consequently is a dwarf and displays a shorter hypocotyl than the wild type [41,42], clearly revealed that A1 treatment was not able to rescue the hypocotyl elongation phenotype, thus pointing out that A1 requires both GA and DELLA for its growth-promoting effect. Surprisingly, the addition of the A1 compound could not rescue the hypocotyl phenotype of the GA receptors’ *gid* double mutants, suggesting that this small compound acts upstream of the GA receptors [38]. Therefore, although this work presents an interesting compound able to increase plant growth, there is still a need to investigate the MoA (mode of action) of the A1 small molecule to make it a remarkable agrochemical candidate.

Totipotency and pluripotency of plant cells are the basis of plant regeneration; indeed, plant cells can be totipotent or pluripotent, depending on whether they are able to differentiate into a complete individual or into a specific group of tissues or organs from one cell [43]. Regeneration studies in angiosperms have gained popularity with the advent of tissue cultures and biotechnological breeding procedures [44]. The main steps in shoot regeneration from tissue explants are the acquisition of competence and the commitment to form shoots; achievement of the process requires incubation in an auxin-rich callus-induction medium followed by incubation in a cytokinin-rich shoot-induction medium [45,46].

In Arabidopsis, auxin treatment triggers tissue proliferation resembling premature roots and leads to a high, local increased expression of the CK (cytokinin) receptor AHK4 (ARABIDOPSIS HISTIDINE KINASE 4) encoding gene. Then, CK treatment converts early roots into shoots, mainly in those regions where expression of the CK receptor genes is induced [46,47]. Although the regeneration process offers promising approaches to studying cellular reprogramming and developmental plasticity, there are still several plant species that are recalcitrant. Therefore, besides natural plant hormones, a wide range of PGRs (plant growth regulators) have been tested and employed to optimise tissue cultures. With the aim of isolating new compounds able to trigger shoot regeneration, Motte et al. [48] performed an HTS (High Throughput Screening) on root explants of the GAL4-GFP enhancer trap line M0167 [49], which displays expression of the shoot marker *LSH4 (LIGHT-DEPENDENT SHORT HYPOCOTYLS4*) [48]. Screening of a library of 10,000 small molecules revealed only one compound, Phe-Ade (*N*-phenyl-9*H*-purin-6-amine) (Figure 3), able to induce *LSH4* expression and consequently shoot formation. Interestingly, Phe-Ade had been previously described as a CK [50], although less active in CK bioassays. The transcriptomic profile of plants treated with Phe-Ade or the adenine-based cytokinin 2-iP (2-isopentenyladenine) is quite similar, suggesting that Phe-Ade functions as a CK-like molecule and undergoes an akin CK homeostasis process. Interestingly, Phe-Ade requires CK signalling, which is mediated by its direct binding to the CK receptor AHK4, albeit at high concentrations, which seems likely to be contradictory to the strong regenerative ability displayed by the Phe-Ade compound. One explanation is that this small molecule inhibits the activity of the CKX (CYTOKININ OXIDASE/DEHYDROGENASE) enzymes, as shown by in vitro assays. This results in an accumulation of endogenous CK levels and enhanced CK signalling, without the drawback of side effects such as cytotoxicity and inhibited shoot growth. In addition, the Phe-Ade compound only moderately induces callus growth, an aspect which is highly eligible in tissue culture [51]. This study allows us to emphasize the importance of HTS, an extraordinary tool for repurposing already-categorised compounds into new applications, allowing us in this case to identify a promising compound to further explore and optimise tissue culture methodologies.

Although the role of exogenous auxin to optimise callus and shoot formation has been extensively examined [51], little is known about the impact of the endogenously produced auxin in shoot regeneration. A recent study investigated the effects of commercially available auxin biosynthesis and transport inhibitors in this process. Two IAA biosynthesis inhibitors, PPBo (4-phenoxyphenylboronic acid) and Kyn (L-kynurenine), and two inhibitors of auxin polar transport, NPA (1-naphthylphthalamic acid) and TIBA (2,3,5-triiodobenzoic acid), were evaluated (Figure 3). Results showed that treatment with these molecules in a CIM (callus-inducing culture) containing exogenous auxin suppresses callus growth, while shoot regeneration is significantly enriched in a SIM (shoot-inducing culture). Interestingly, this was not due to a reduced amount of endogenous auxin, as revealed by an analysis of the auxin-responsive reporter *DR5::GUS* [52]. Indeed, treated explants showed a greater and more consistent auxin response compared to the control explants, thus suggesting an altered spatiotemporal auxin distribution, which could trigger the shoot regeneration competence. The expression of RAM (Root Apical Meristem) stem cell niche establishment-related genes, which are involved in auxin-mediated shoot regeneration process [53], were consistently upregulated in treated plants. In particular, the expression level of *PLT7* (*PLETHORA7*) was strikingly increased by treatment with both PPBo and NPA, implicating this gene as a promising candidate for future development in this field [54].

Nowadays, the concept of overpopulation is one of the causes of environmental issues and the constant decline of natural resources. Therefore, new biological resources for renewable fuel alternatives are needed. Plant biomass is an attractive bioresource, mainly stored as biopolymers in plant cell walls; in fact, cellulose and hemicellulose polysaccharides are sugar resources for biofuels and other biomaterials [55]. By means of a chemical genetic strategy, the small molecule LS (lasalocid sodium) (Figure 3) has been identified for its ability to alter plant cell wall traits in tobacco BY-2 cells [56]. LS is a carboxylic acid ionophore that binds divalent and monovalent cations, already known as an antibacterial and an anti-coccidial agent [57,58], whose activity and effectiveness have never been assessed in plants. LS was shown to increase cell wall thickness and saccharification efficiency, while not significantly affecting its sugar composition. The effect of LS *in planta* was assessed on Arabidopsis seedling; treatment with LS on etiolated seedlings resulted in reduced hypocotyl length and open cotyledons. Since cell wall loosening affects cell morphology and in turn cell elongation, it seems likely that LS treatment results in an increase in cell wall looseness, both in cell culture and in plants. Consistently, data from a microarray analysis showed that cell-wall-related genes were among the upregulated categories from GO (Gene Ontology) analysis, and the class III Prxs (peroxidase) and JA (jasmonic acid)-related genes were among the significantly upregulated genes in the LS treatment samples. Interestingly, both Prxs and the integrity of the plant cell wall are known to be involved in cell elongation and biotic stress response, which induces JA signalling [59,60]. Prxs activity on cell elongation occurs with the increase in ROS (reactive oxygen species). Interestingly, accumulation of ROS and increased JA levels were associated with plant cell wall damage [61]. Thus, the authors successfully identified a new compound able to alter plant cell morphology through an increase in the enzymatic saccharification efficiency of cell walls materials, proposing LS as a chemical biological approach for cell wall engineering as well for the investigation of cell wall damage responses, but also to develop new strategies for biomass applications [56].

The gaseous phytohormone ethylene is produced in seedlings grown in darkness to increase their soil penetration ability, and it induces reduced cell elongation, radial expansion, and the apical hook. These phenotypes are part of the so-called triple response typical of hypogeic plants, which allows for more rigidity and protection of the shoot apex [62]. This hormone has also been implicated in flowering, fruit development and ripening, notably important agronomy traits. As such, great efforts have been made to develop technologies for the manipulation of ethylene levels in plant tissues by using chemicals: One example is the 2-chloroethylphosphonic acid (ethephon), commercially available and used to promote fruit ripening, abscission, and flower induction [63,64]. Oh et al. [65], aiming to discover new non-gaseous chemicals with ethylene-like activity, screened a set of 9600 chemicals using a triple-response bioassay on dark-grown Arabidopsis seedlings [65]. The compound EH-1 (*N*-[(1,3,5-trimethyl-1*H*-pyrazol-4-yl)methyl]-*N*-methyl-2-naphthalenesulfonamide) (Figure 3), a sulfonamide derivative, was able to induce short hypocotyls and an exceeding apical hook. To further assess EH-1 biological activity, a set of structural analogues has been designed by replacing the naphthalene moiety with a phenyl moiety containing different substituents and substitution patterns. Among these compounds, several new molecules displayed promising activity on the apical hook formation and root and hypocotyl elongation. A transcriptomic analysis of EH-1-treated plants compared with plants treated with ACC (1-aminocyclopropane-1-carboxylate), a precursor of ethylene biosynthesis, showed that ACC treatment triggers ethylene-responsive marker genes such as extensins and peroxidases, while expression of these genes was not affected in EH-1-treated plants. These results revealed that EH-1 treatment alters gene expression differently from ACC, as only a few of the differentially expressed genes have this in common, thus withholding the hypothesis that EH-1′s effects could be mediated by triggering ethylene synthesis or signalling [65]. Unveiling the mechanism of action of EH-1 will lead to insights into the action of ethylene and the ethylene-mediated triple response, while it could become a promising compound to improve the fruit yield of crop species.

**Figure 3 plants-12-01729-f003:**
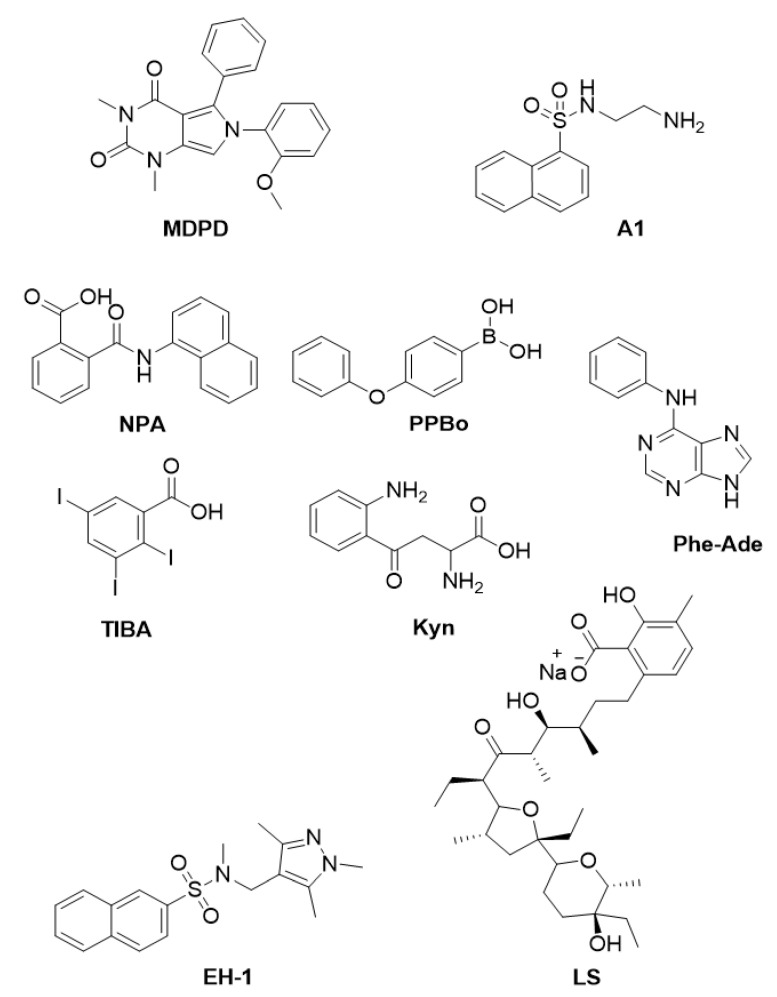
Chemical structures of small molecules affecting plant development. MDPD [32], A1 [38], LS [56], EH-1 [65], Phe-Ade [48], PPBo, NPA, TIBA and Kyn [54].

## 4. Small Molecules to Find Out How Plants Cope with Biotic and Abiotic Stresses

The world population is predicted to grow to up to 9.6–12.3 billion people by the year 2100 [66]. This in turn will imply a high demand for agricultural products, which should increase by about 50% between 2010 and 2050 [67]. Besides that, the increasing warning from global climate change and the concern about the future survival of crops has prompted researchers both to deepen the study of how plants face abiotic and biotic stress and to develop strategies to improve plant tolerance to abiotic stress as well as plant defense from pathogens. As for the latter, recently, two noteworthy studies based on a chemical genetic strategy have been published; the first identified three components of the methylosome as negative regulators of disease resistance [68], while the other isolated small molecules to inhibit the FERONIA RLKs (receptor-like kinases), which negatively regulate root immunity and plant response to soil-borne diseases [69]. To investigate how the cytosolic NLRs (Nucleotide-binding leucine-rich repeat) immune receptors elicit the ETI (effector-triggered immunity), Huang et al. [68] performed an HTS (13,600 commercially available compounds) aimed at identifying small molecules able to suppress the autoimmune phenotype of the gain-of-function *chs3-2D* (*chilling sensitive 3, 2D*) mutant [68,70]. It was identified that one compound, Ro 8-4304, (4-[3-[4-(4-fluorophenyl)-1,2,3,6-tetrahydro-1(2*H*)-pyridinyl]-2-hydroxypropoxy] benzamide hydrochloride) (Figure 4), was shown to specifically repress overexpression of *PR1* and *2* (*PATHOGEN RELATED*) genes in the mutant background. By screening an EMS-mutagenized *chs3-2D* mutant population for insensitivity to Ro 8-4304, the *Aticln* (I, current; Cl, chloride; n, nucleotide) knock-out mutant was isolated. AtICln is a component of the methylosome complex with SmD3b (SMALL NUCLEAR RIBONUCLEOPROTEIN D3B) and PRMT5 (PROTEIN ARGININE METHYLTRANSFERASE 5), and it functions as a chaperone to ensure the proper spliceosome assembly [71]. Consistently, the *chs3-2Dsmd3b* double mutant is insensitive to the Ro 8-4304 compound, while *chs3-2Dprmt5* is lethal. Although the methylosome complex is not likely to control the alternative splicing of the *CHS3* transcript, these data indicate that splicing is involved in the negative control of plant immunity for snRNPs (small nuclear ribonucleoproteins) biogenesis [68]. The identification of the molecular targets of the Ro 8-4304 compound will allow us to shed light on how the methylosome is involved in plant immunity.

Although soil-derived plant pathogens impair plant growth and crop yield, the root immune response has not been deeply studied compared to leaf immunity due to the variability of the soil environment. Nevertheless, it is well established that FER is involved in the response to both biotic and environmental stress [72,73,74,75,76,77]. FER also plays a function in plant developmental processes such as hypocotyl and root growth and in hormonal growth control, as for brassinosteroids and auxin [78,79]. To further corroborate the crucial role of FER in plant development and response to stresses, a model based on the Chinese theory of yin-yang balance has recently been presented, in which TMKs (Trans-Membrane kinases) and FER finely control the acidification–alkalization balance and the abscisic acid (ABA)–Auxin balance. The crosstalk between TMKs and FER, mediated by ABI2 phosphorylation, directs the plant towards growth or alternatively the response to stress by varying the ABA/Auxin ratio and, in turn, alkalization/acidification [80]. Given the prominence of containing soil-borne disease to improve crop productivity, a screening was conducted to identify small molecules capable of inhibiting FER kinase activity in vitro and *in vivo*, on different crops, namely tobacco, tomato, and rice [69]. Based on the effectiveness of inhibiting FER autophosphorylation activity *in vitro*, and based on their specificity compared to the other three kinase families, four small molecules were selected, of which staurosporine and lavendustin A are natural molecules [81,82], while reversine and cenisertib were previously synthesized as Aurora A kinase inhibitors and anti-cancer molecules (Figure 4) [83,84]. Although these small molecules are likely to present off-targets, as suggested by the contrasting effects on root growth [69], they can represent interesting lead compounds to develop new and more selective inhibitory compounds. Furthermore, any effects on leave diseases not assessed in this research should also be investigated.

The phytohormone ABA is the key hormone in a plant’s response and adaptation to environmental cues and stress [85]. The core ABA signalling pathway has been well characterized (Finkelstein and Rock, 2002). Once ABA binds to one of its receptors belonging to the family of PYR1 (PYRABACTIN RESISTANCE 1), PYL (PYR1-like) proteins, or RCARs (REGULATORY COMPONENT of ABA RECEPTORS), the downstream elements PP2Cs (clade A type 2C protein phosphatases) are inhibited and, in turn, the restraint of the downstream positive elements SnRK2 kinases (SNF1-related kinases) is released, and the ABA-mediated stress response pathway is activated [85]. Several structural studies have contributed to point out the ABA receptor binding and the ABA signalling mechanism, defining a gate–latch–lock model for the perception and transduction of ABA [86,87,88,89,90]. This allowed the development of pharmacological approaches that aim at isolating ABA-mimicking small molecules, which, conversely to ABA, would be more stable both as a molecule and *in planta*. AM1 (ABA Mimic 1), a sulfonamide compound, has been isolated as an ABA agonist, able to trigger the interaction between a His-tagged PYR1 receptor and a biotin-tagged HAB1 (HYPERSENSITIVE TO ABA1), one of the Arabidopsis PP2C. By means of a yeast two-hybrid screen, it was shown that AM1 binds all the ABA receptors except PYL4 and PYL6; consistently, AM1, once bound to these ABA receptors, is able to inhibit HAB1 phosphatase activity [91]. Treatment with AM1 induces the same transcriptional response than ABA, as proven by a genome-wide analysis. Consistent with its structural and binding features, AM1 affects plant development and response to abiotic stress similarly to ABA. Indeed, wild-type AM1-treated seeds did not germinate as ABA-treated ones, while AM1-treated plants were more tolerant to drought stress [91]. Although further studies on the pharmacokinetics and toxicity of AM1 are needed, this could prove to be a compound of considerable interest to increase the tolerance of crops to abiotic stresses. Indeed, unlike ABA, it is a very stable molecule and, unlike previously isolated compounds, namely PB (pyrabactin) [89], it is effective for several ABA-mediated processes. PB, which is an ABA receptor agonist, has been shown to be predominantly effective in seeds, where its receptor PYR1 is most highly expressed [89]. To identify ABA receptor agonists that also affect the drought response, a new molecule has been rationally designed, CB (cyanabactin) (Figure 4), based on the chemical structure of ABA [92]. CB preferentially activates PYR1 and its paralog PYL1, both belonging to the IIIA ABA receptor subfamily, with low nanomolar activity. An X-ray crystallographic structure of CB in a complex with PYR1 illustrates that cyanabactin’s arylnitrile mimics ABA’s cyclohexenone oxygen and engages the tryptophan lock, a key component required to stabilize activated receptors. The sulfonamide and 4-methylbenzyl moieties mimic ABA’s carboxylate and C6 methyl groups, respectively. Expression analysis of ABA-responsive genes, namely *MAPKKK18* (*MITOGEN-ACTIVATED PROTEIN KINASE KINASE KINASE 18*), *RAB18* (*RAS-RELATED PROTEIN18*), and *RD29B* (*RESPONSIVE TO DESICCATION 29B*), clearly revealed that treatment with CB strongly induced expression in a wild-type background, while in a *pyr1pyl1* double mutant background, expression of the ABA-responsive genes was significantly decreased [92] in contrast to ABA treatment, due to the redundancy of ABA receptors. Since ABA-dependent plant tolerance to drought is, at least in part, mediated by ABA’s control of stomata closure, the authors measured whole-plant stomatal conductance to verify the *in planta* effect of CB treatment [92]. Results from these analyses demonstrated that CB treatment reduces stomatal conductance and, in turn, increases leaf temperature, thus suggesting the activation of the ABA-mediated response [92]. Therefore, this compound has been proven to mimic ABA-mediated responses such as inhibition of seed germination, transcriptional induction, and reduced plant transpiration as a result of the activation of ABA receptors of the IIIA subfamily.

The number and size of stomata are dependent on both internal inputs, such as hormones, and environmental clues, such as light, water availability, temperature [93,94,95]. A high stomatal density increases transpiration and promotes photosynthesis, while a reduced number of stomata per leaf area allows optimal water-use efficiency [96]. Thus, under drought stress conditions, a low stomata number can help the plant to cope with the stress, although it will result in a decreased crop yield. However, since stomatal development is considered to play a key role in crop plant productivity and water-use efficiency, reducing stomatal closure is one of the primary strategies for improving crop yields under drought conditions. Bikinin was identified by a HTS from a commercial library (ChemBridge) as the first nonsteroid molecule affecting stomatal development [97]. Later, bubblin was identified by HTS on Arabidopsis seedlings and perceived to modulate stomatal development [98]. Indeed, bubblin was shown to promote an increase in stomatal density combined with a severe inhibition of seedlings development, thus preventing its use for crop improvement in arid soils. This compound is a pyridine-thiazole derivative, which inspired the synthesis of five derivatives (A1 to A5). SAR (structure–activity relationship) analysis of the bubblin derivatives revealed that the pyridine ring of bubblin is essential for its activity as compound A2, where the pyridine ring was replaced by a phenyl ring and was inactive. Likewise, the elimination of the bromine atom on the aromatic ring in the four positions of the thiazole core (A3) also led to loss of activity. Moreover, compounds A4 and A5, wherein the halogen was replaced with a methyl group (A4) or methoxygroup (A5), showed no activity, suggesting a key role of the bromine atom for effectiveness in stomata clustering [98]. It is noteworthy that the compounds with stomatal clustering activity inhibited seedling growth in a dose-dependent manner and vice versa. With the aim of identifying compounds able to trigger stomatal developmental but with no effect on plant growth, Ziadi et al. [99] performed a plant-based phenotypic screening. CL1 and CL2 were identified as stomatal development-enhancing molecules, although treatment with both these compounds resulted in seedlings’ growth defects at high concentrations. Taking into account previous experiences as prior-art, their strategy was to develop new compounds able to increase the number of stomata while lessening the toxicity at high concentrations. Preliminary SAR studies were performed, and analogues ZA155, lacking a trifluoromethyl (CF_3_) group in the C3-position, or ZA099, lacking the aryl group in the C5-position of the pyrazole core, were synthesized. Since treatment with ZA155 caused growth-inhibition effects while ZA099 did not, this latter was selected as a hit compound. Eight new derivatives were designed primarily by introducing various substituents featuring various electronic and steric properties on the aryl group in the C5-position of the pyrazole moiety, while maintaining the CF_3_ group in the C3 position. The effectiveness of these compounds was evaluated in terms of inhibition of seedling growth and number of stomata. Only one of these small molecules, ZA144 (Figure 4), has been proven to be able to increase the number of stomata without affecting plant development [99]. Analysis of the dry weight corroborated this proof, confirming the ZA144 compound as a promising candidate for further studies, which will allow us to identify the cellular target/s of this small molecule.

One of the most worrying environmental stresses, which severely compromises crop growth and productivity, is salt stress, which is likely to become increasingly pervasive due to water depletion in soil. Therefore, identifying small molecules able to increase salt stress tolerance in plants is a compelling goal. Recently, a compound able to increase salt stress tolerance in Arabidopsis seedlings has been identified through a chemical priming approach [100]. Priming is a plant adaptive process that allows a faster and more efficient response against future stresses [101]. Treatment with this small molecule, Natolen128 (NaCl tolerance enhancer128), (*N*-[3-(2-oxo-1-pyrrolidinyl)phenyl]-spiro[bicyclo[3.2.1]octane-8,20-[1,3]dithiolane]-3-carboxamide), was performed on 4-day-old grown wild-type seedlings 24 h before the salt stress (100 mM NaCl). As a negative control, a compound with a chemical structure (*N*-[4-(1,1-dioxido-2-isothiazolidinyl)phenyl]-spiro[bicyclo[3.2.1]octane-8,20-[1,3]dithiolane]-3-carboxamide) similar to Natolen128 was isolated, but with no effect on salt tolerance—Necolen124. The survival rate of the seedlings under salt stress for 4 days was higher than 80% for plants primed with Natolen128, while it was equal to 0 for Necolen124- and mock-treated plants. A transcriptomic analysis of Natolen128-treated seedlings compared with Necolen124-treated seedlings and with or without salt-stress revealed that priming with Natolen128 triggers the expression of ethylene biosynthetic genes as well as of *AtPGB1* (*Hemoglobin/Phytoglobin1*), which is known to modulate NO (Nitric Oxid) level [102]. Ethylene is involved in the response to salinity stress [103], thus suggesting that Natolen128 might increase ethylene level to prime the stress response. Treatment with Natolen128 induces the expression of several hypoxia-responsive genes, and ethylene is produced during the early response to hypoxia conditions, triggering *AtPGB1* expression, thus suggesting that priming with this small molecule could result in a O_2_ deprivation-like status [100]. Interestingly, the authors had previously isolated another compound, FSL0260, as an inhibitor of mitochondrial complex I, which in turn caused a decrease in ROS accumulation and an increase in salt stress tolerance [104]. The understanding of the action mechanisms of these promising molecules is still lacking, and it is needed for their future application to improve the tolerance of crop plants to salt stress.

**Figure 4 plants-12-01729-f004:**
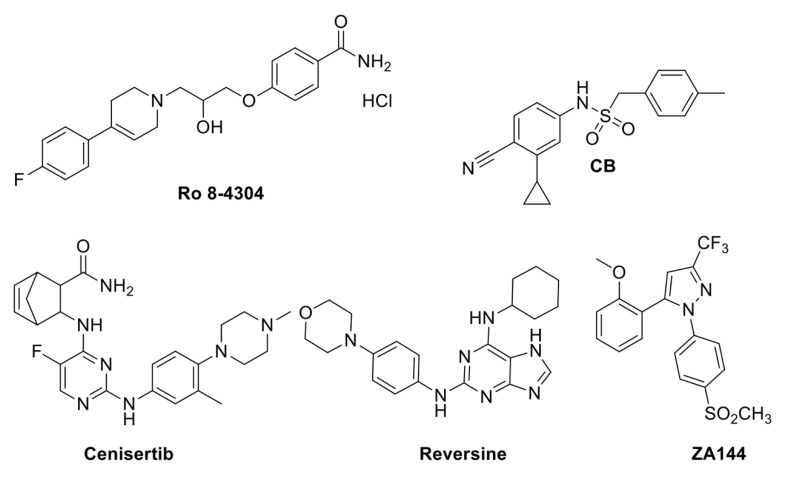
Chemical structures of small molecules involved in plant response to biotic and abiotic stress. Biotic stress: Ro 8-4304 [68], Reversine and cenisertib [69]. Abiotic stress: CB [92], ZA144 [98].

## 5. Small Molecules to Study Epigenetic-Controlled Processes

Plants have different levels of gene expression control and several intricate mechanisms to properly carry out cellular division, development, and growth. At the epigenetic level, the extent of chromatin condensation determines the accessibility of the transcription machinery to the genome, thus modulating gene expression without DNA mutation. Two main epigenetic processes are involved in this regulatory mechanism: DNA methylation and histones modifications. Specific enzymes, named “writers”, catalyse these changes on the DNA and chromatin, while the “erasers” catalyse their elimination, allowing the epigenetic state to be, at different extents, dynamic and reversible. Small molecules can alter these regulatory mechanisms, thus modulating the epigenetic landscape and in turn affecting cell differentiation/proliferation, morphogenetic processes, developmental phases, and plant response to the environment.

In plants, chemical strategies applied to epigenetic studies often focus on a specific epigenetic marker or mechanism and utilize small molecules, usually inhibitors, which are already available. Indeed, due to the high conservation of epigenetic factors between animals and plants, plant chemical epigenetics often exploit compounds previously proven to be effective against mammalian targets, an approach referred to as small molecule repurposing.

DNA methylation consists of the addition of a methyl group on the pyrimidine ring of cytosine in DNA (5-mC). Cytosine methylation has been associated with gene expression control, the silencing of transposable elements and the maintenance of genome integrity, imprinting and transgenerational memory, plasticity, and adaptation to stress [105,106]. In plants, several molecules have been widely applied to directly inhibit DNMTs (DNA methyltransferases) activity or indirectly reduce DNA methylation. The cytosine analogues 5-azaC (5-azacytidine) and zebularine (Figure 5) are the most applied DNMTs inhibitors [1]. Chemical analogues of cytosine, incorporated in the DNA, can form covalent adducts with DNMTs, mediating enzyme degradation and in turn leading to a genome-wide reduction in DNA methylation [1,107].

In tissue cultures, chemical treatment has great effects on phenomena based on reprogramming, proliferation and (re)differentiation, such as SE (somatic embryogenesis), ME (microspore embryogenesis), and shoot regeneration [108,109,110,111]. These in vitro processes are useful as agriculture biotechnological tools as well as models to investigate molecular and physiological processes occurring during plant developmental phases [112]. The chemical approach is largely dependent on the plant species as well as on the concentration of the compound and the span of the treatment. For instance, in *Brassica napus* and *Hordeum vulgare*, 4 days of treatment with 5-azaC improved embryo induction, while longer treatment resulted in a reduction of developed embryos [113]. In *Coffea canephora,* it was reported that treatment with 5-azaC decreased the expression level of *LEC1* (*LEAFY COTYLEDON1*) and *BBM* (*BABY BOOM*), hindering the embryogenic program, while treatment 35 days after embryogenic induction improved SE [114]. Recently, a study on the role of DNA methylation on SE in *Arabidopsis* using 5-azaC revealed an inverse correlation between the global methylation level (reduced) and the expression level of genes encoding DNA methylases/demethylases, increased and decreased, respectively. In addition, SE regulatory genes were differentially expressed between 5-azaC-treated and methylase mutant cultures. Indeed, in agreement with [114], *LEC2* (*LEAFY COTYLEDON2*)*, LEC1* and *BBM* genes were downregulated in 5-azaC-treated cultures, while the same genes were upregulated in mutant cultures lacking combinations of DNA methylases, with improvement of embryogenic induction [115]. By trapping the DNA methyltransferase, 5-AzaC causes passive DNA demethylation and, thus, a global and random reduction of the 5-mC level [116]. In contrast to 5-AzaC, mutations in *CMT3* (*CHROMOMETHYLASE3*) and *DRMs* (*DOMAINS REARRANGED METHYLTRANSFERASEs*) result in a sequence-specific decrease in DNA methylation, while the overall methylation is less affected [115].

It has been established for a long time that stresses can trigger genetic variation, inducing the activation of TEs (transposable elements) [117,118,119]. DNA methylation plays an important role in the control of transposon activity in plants. Therefore, the chemical inhibition of DNA methyltransferases has been utilized to study the effect of DNA methylation on TE silencing as well as the effects of TEs activation on plants. Several studies reported that 5-azaC or zebularine treatment triggers TE activation [120,121,122]. A recent study focuses on DNA methylation in the mobilization of the retrotransposon *ONSEN.* Usually, transgenerational transposition of *ONSEN* is too rare to be detected [123], while, following heat stress, *ONSEN* is transcribed, and a full-length extrachromosomal *DNA* (ecDNA) is generated [124]. Indeed, this Ty1/copia-like retrotransposon has heat-responsive elements in its LTRs, recognized by heat-responsive plant transcription factors under high temperature [121,125,126]. Thieme et al. [125] developed a method to amplify *ONSEN* mobilization and overcome the low transposition activity of TEs in natural accessions. Indeed, they used the simultaneous combined chemical inhibition of DNA methyltransferases with zebularine and RNA polymerase II with α-amanitin under heat shock stress. This method, named BUNGEE (breeding using jumping genes), increased *ONSEN* mobility, and new copies were integrated and inherited. The *ONSEN* high copy TE lines (hcLines) stably maintained copy number over three generations [125]. Among these lines, the authors found a single novel insertion of *ONSEN,* resulting in a loss-of-function of *RPI2*(*RIBOSE-5-PHOSPHATE ISOMERASE 2*), which showed increased tolerance to drought [126]. It could be concluded that chemical TE mobility induction is a powerful tool for enforcing crop improvement to face the ongoing climate change, being also useful for investigating plant plasticity and adaptation mechanisms to the environment.

Small molecules active against histones’ Post Translational Modifications (PTMs) have been widely developed, with histone deacetylase inhibitors being the most applied in plants. HATs (Histone acetyltransferases) and HDACs (Histone deacetylases) are among the best structurally characterized enzymes of histone modifications. Their activity is related to gene activation and repression, respectively, as the acetylation of histone tails relaxes chromatin by neutralizing the positive charge of lysine interacting with the phosphate backbone of DNA. The HDAC inhibitors TSA (trichostatin A) (Figure 5) and NaB (sodium butyrate) have been used for a long time, and as for DNA methylation inhibitors, they have been largely applied in plant cultures.

Since the compelling necessity to improve plant ability to adverse environmental and new climatic conditions, TSA and NaB have great applicability for in vitro regeneration. Chemical treatment combined with abiotic stress, as cold or salinity stress, largely modify the reprogramming efficacy. Currently, several promising strategies for SE and ME induction are based on TSA in Brassica, barley, and particularly in wheat [127,128].

By screening several HDACs inhibitors, the Ky-2 HDAC inhibitor has been selected due to its effectiveness in enhancing tolerance to salt stress in Arabidopsis [129]. At the molecular level, Ky-2 treatment resulted in increased global histone acetylation level and upregulation of SOS*1* and *SOS3* (*SALT OVERLY SENSITIVE 1 and 3*), which respectively encode a Na^+^/H^+^ antiporter and an EF-hand Ca^2+^-binding protein, consistent with the increased tolerance to salt stress. Accordingly, treatment with Ky-2 resulted in the enrichment of the histone H4 acetylation mark in the *SOS1* and *SOS3* loci and in the decrease in the [Na]^+^ level. Moreover, Ky-2 treatment caused an accumulation of stress-related metabolites such as proline and polyamines [129]. In addition, the chlamydocin analogs Ky-9 and Ky-72, which share a common cyclic tetrapeptide with differing side chains, enhanced high-salinity stress tolerance in *A. thaliana* [130]. Chemical-treated plants showed an upregulation of several salt-responsive genes, but, unlike Ky2-treated plants, not of *SOS1* and *SOS3*. Instead, treatment with both these compounds caused an increase in the expression level of genes related to the anthocyanin biosynthesis pathway *PNP-A* (*Plant Natriuretic Peptide A*), encoding a small peptide of solute homeostasis [131,132], and *OSM34* (*Osmotin-like protein* 34), which encodes an osmoprotectant [133]. Although the MoA of these compounds is still unknown, these results suggest that the HDAC inhibitors have distinct targets and mechanisms for enhancing high-salinity stress tolerance, thus implying that different HDACs are likely to control diverse genes in the stress response pathways [130]. While several HDAC inhibitors have been widely applied in plants, few cases of HATs inhibition have been reported so far. *Arabidopsis* has five HAT proteins related to the yeast GCN5 (*GENERAL CONTROL NON-REPRESSIBLE 5*), the HAG1-5 (HISTONE ACETYLTRANSFERASE OF THE GNAT/MYST SUPERFAMILY 1-5). The GCN5 inhibition activity in plants by the molecule MB-3 (α-methylene-γ-butyrolactone) was recently reported. MB-3 is the first inhibitor, which efficiently targets the catalytic active site of mammalian GCN5 [134]. As the targeted amino acids are conserved in the *Arabidopsis* homologous, it was assumed that MB-3 was active in plant cells [135]. The compound was evaluated in protoplasts [136] and in seedlings [137]. Consistent with GCN5 inhibition, MB-3-treated *Arabidopsis* seedlings showed a reduced amount of H3K4ac and H3K9ac and displayed a significant reduction of root growth and chlorotic leaves. Expression of GCN5 target genes involved in abiotic stress response, namely *NRT1.5* (*NITRATE TRANSPORTER* 1.5, *PROT1* (*PROLINE TRANSPORTER 1*), *TPK5* (*TWO-PORE POTASSIUM CHANNEL* 5), *COR6.6* (*COLD REGULATED 6.6*), was significantly downregulated after a 3 h treatment, while it was restored to a normal level after 24 h [135]. Interestingly, GNC5 target genes involved in developmental processes, namely *SPL3* and *9* (*SQUAMOSA PROMOTER BINDING PROTEIN-LIKE 3* and *9*), were not affected by MB-3 treatment, possibly due to further levels of control, namely through miRNAs [138].

HACs (Histone Acetyltransferases) proteins, homologs of the p300/CBP human family, represent another subfamily of *Arabidopsis*. Taking advantage of the homology between the human p300/CBP and the *Arabidopsis* HAC catalytic domains, previously described human p300/CBP inhibitors [139,140] have been tested for their efficacy through a phenotypic screening on Arabidopsis dark-grown seedlings. Among them, six cinnamoyl derivatives [141] and three newly synthesized structural analogues were selected, as they were able to phenocopy the phenotype of *hac* mutants, namely a shorter hypocotyl under dark-growth conditions. The efficacy of the most promising compound, 2,6-bis(3-bromo-4-hydroxybenzyl)cyclohexanone (2c) (Figure 5), was further assessed at the molecular level by demonstrating that treatment with 2c caused a significant decrease in the expression level of four *HAC* target genes, namely *HLS1* (*HOOKLESS1*), *ORG1* (*OBP3 RESPONSIVE GENE1*), *LTP5* (*LIPID TRANSFER PROTEIN 5*). To corroborate these results and prove the specificity of compound 2c, the catalytic domain of HAC1 was expressed and purified to run an in vitro assay, which revealed that compound 2c was able to reduce HAC1 activity by 83% compared to the control [141]. These results substantiate the chemical strategy as a useful tool in plant epigenetic studies, but on the other hand, they suggest plants as a useful tool to isolate and validate new small molecules.

In contrast to DNA methylation and histone acetylation, the chemical modulation of histone methylation in plants has few examples reported so far, probably due to the complexity of this epigenetic mark. Indeed, methylation can occur at two different residues, K (lysine) or R (arginine), in diverse sites of the histones tails and with one to three methyl groups, thus resulting in an opposite chromatin condensation level and in turn transcriptional effect. In plants, several HKMT (histone lysine methyl transferases) have been identified by their homology with the animal SET domain [142,143]. So far, just two molecules have been proven effective against plant HKMTs: 1,5-bis-(3-bromo-4-methoxyphenyl)penta-1,4-dien-3-one (RDS 3434), as an inhibitor of PRC2 SET domain activity in Arabidopsis [144], and BIX-01294, a diazepin-quinazolin-amine derivative, as a SUVR4 HKMT inhibitor in *Brassica napus* and *Hordeum vulgare* [145].

PRC2 (POLYCOMB REPRESSIVE COMPLEX 2) is a repressive complex highly conserved between animals and plants, which catalyses the methylation of H3K27 thus repressing the expression of a plethora of developmental genes. In Arabidopsis, the PRC2 methyltransferase subunit is encoded by three homologs: CLF, MEA and SWN (CURLY LEAF, MEDEA and SWINGER), which share a highly conserved SET domain and have partially redundant roles [146,147]. RDS 3434 (Figure 5) has been synthesized as a human PRC2 inhibitor active against the catalytic subunit EZH2, and its effectiveness has been proven on the oncogenic human monocyte cell line U937 [139]. Recently, RDS 3434’s effectiveness against *Arabidopsis* EZH2 during the seed to seedling transition stage was assessed and proven. Indeed, treatment with RDS 3434 results in a dose-dependent decrease in the H3K27me3 bulk proteins of wild-type seedlings and an increase in the expression level of two PRC2 target genes, namely *DAG1* (*DOF AFFECTING GERMINATION 1*) and *WRKY70* (*WRKY DNA-BINDING PROTEIN 70*). Consistently, H3K27me3 enrichment was decreased in these target genes’ loci. The specificity of RDS 3434 was proven by demonstrating an effect of the inhibitor on other methylation marks, such as H3K4me3 and H3K36me3, and on non-target genes of PRC2.

The *clf-28 swn-4* double mutation results in a delayed germination compared to the wild-type seeds [148]. A reduction in PRC2 activity by RDS 3434 treatment affected both seed germination and root apical meristem, consistent with the previously described phenotypes of the *clf-29* and *clf-28 swn-7* mutants, which lack one or two PRC2 catalytic subunits [149]. These promising findings led to the conclusion that RDS 3434 is also an effective inhibitor in plants, thus representing a powerful tool to further investigate PRC2 roles.

BIX-01294 (Figure 5) was previously identified as a very specific inhibitor of G9a histone lysine methyltransferase [150,151], able to reduce H3K9me2 levels in several mammalian cell lines [152,153,154,155,156]. Berenguer et al. [145] has recently repurposed this compound in plants, investigating its effects on ME processes for two main reasons: (i) ME is a process based on cell reprogramming, (ii) plants’ SUVR4-like HKMTs are highly related to the animal G9a HKMT, which has a crucial role in H3K9 methylation during mammal embryogenesis [150]. In vitro ME is normally induced by stress and widely used in plant breeding for the rapid production of DHs (doubled haploids), but its regulatory mechanisms are largely unknown. To unveil these mechanisms, the authors analysed changes in H3K9 methylation in rapeseed and barley, using in vivo gametophytic development for comparison [145]. H3K9 methylation level increased during the gametophytic program progression, while, following stress induction, ME initiation was characterized by lower H3K9 methylation levels, increasing later as embryo differentiation proceeded. Therefore, BIX-01294 was tested for the efficiency of microspore reprogramming to embryogenesis. Short-term BIX-01294 resulted in a significant reduction in H3K9 methylation level and a consequent increase in microspore embryogenesis induction. Moreover, since 5-azaC has also been reported to promote microspore reprogramming and embryogenesis induction in rapeseed and barley [113], the DNA methylation state following chemical treatment was analysed. Indeed, BIX-01294 treatment caused a decrease in global DNA methylation, thus suggesting that microspore reprogramming and totipotency acquisition require a release of chromatin condensation through a transient decrease in global repressive marks, as H3K9me2 and DNA methylation. Interestingly, as for 5-azaC studies, timing of the treatment is crucial. Indeed, when BIX-01294 was supplied in the late stages of microspore embryogenesis, it hindered development and blocked the process at the proembryo stage, indicating that de novo H3K9 methylation is required for the differentiation of embryo cells [145].

**Figure 5 plants-12-01729-f005:**
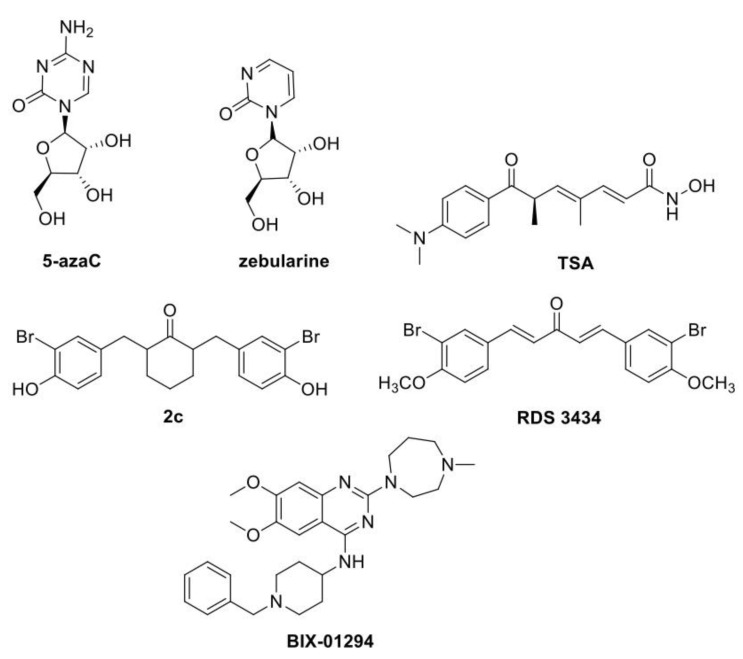
Chemical structures of small molecules involved in epigenetic processes. DNA methylation: 5-azaC [115], zebularine [125,126]. Histone Acetylation: TSA [128], 2c [141]. Histone methylation: RDS 3434 [144], BIX-01294 [145].

## 6. Future Perspectives and Challenges

Although the use of small molecules has been applied successfully for a long time, in plants its potential has been experienced more recently and is moderately becoming a tool in basic plant research. Indeed, chemical compounds can rapidly overcome mutant lethality or plant redundancy in gene families and have the useful benefit of being able to be applied in transient and tunable treatment, allowing a reversible modulation of plant biological systems [19].

In plants, a major challenge in chemical genetic approaches such as the HTS of small molecules libraries is the identification of the target(s) of the small molecule(s) of interest and, thus, the investigation of the MoA. Usually, targets have been identified from screening for mutants insensitive to the selected compound or through a candidate approach, but both methods have limitations [157]. Over the years, other target identification strategies have been developed, including biochemical methods such as affinity purification, or proteomics approaches, which require tagged compounds, SAR studies and even synthetic chemistry. One of these approaches is represented by the “tagged libraries” in which reactive groups are incorporated in small molecules, allowing the simple immobilization of the compounds for their target affinity purification [4,19]. CETSA (cellular thermal shift assay), is another well-suited strategy for identifying direct targets [158], especially at the proteome-wide level. This approach is based on the principle that small molecule–protein interactions can stabilize or destabilize the protein of interest. The protein thermal profile reflects target protein stability and indicates how much is not yet denatured at certain temperatures. Thus, CETSA provides a label-free tool to identify small molecule target proteins [159,160]. However, the plant field still lacks the expertise and technologies of drug discovery [19,161,162]. Indeed, as reported in this review, some compounds, such as A1, EH1 and Ro 8-4304, still lack a clear target and/or the MoA. In contrast, targets of reverse chemical genetic screens are known, although any compounds have yet to be screened in plants, as the screenings are performed in vitro and then tested in vivo to assess their efficacy. Therefore, the real bioavailability and potential ‘off-target’ effects can highly influence these experiments. Specific in vivo assays can be designed for target proteins, relying on labelled inhibitors that react with specific enzymes in an activity-dependent manner. In vivo profiling with these probes consist of a two-step labelling procedure (click-chemistry) [163]. When combined with phenotypic assays, this tool can be utilized to screen probe-derived chemical libraries and find correlations between phenotypes and enzyme activity [164].

An important challenge to be addressed in plant chemical genetics is the compound metabolism *in vivo,* thus including the uptake, bioavailability and toxicity of small molecules [4]. Indeed, the fate of small molecules in plants is still largely unknown. It is assumed that small molecules have generally easy access through the radical system. However, sirtinol derivatives were not effective on solid medium, but they were effective on seedlings in liquid cell cultures, thus suggesting that bioavailability could be higher in other tissues than in the roots [165]. In a recent paper, a specific protocol to measure the uptake of the selected compound inhibitor 2c was designed. Seedling uptake was determined indirectly, measuring the residual amount of compound 2c in the medium by HPLC analysis. According to the results obtained, the plants’ uptake of the inhibitor supplied in the medium was 28.45%, thus indicating that high concentrations can be required to obtain the compound’s efficacy [141]. In addition to the uptake, the conversion of small molecules should also be considered. Indeed, plants have multiple steps of xenobiotics metabolism, including conversion via P450 monooxygenases and conjugation via glutathione transferases. Moreover, several other enzymes are involved in xenobiotics transformation. O-glucosyl and O-malonyl transferases [166] and, more recently, a large gene family of glycosyltransferases were shown to be involved in xenobiotic metabolism in Arabidopsis [167]. In addition, flavin-containing monooxygenases have been recently functionally identified in plants, where they hydroxylate xenobiotics to make them more hydrophilic [168]. Eventually, xenobiotics are compartmentalized into the vacuole, may be secreted into the apoplast, or can be immobilized in the cell wall [4]. It is not atypical for compounds to be chemically transformed, and, for the screens, compounds that are degraded are difficult to be detected. Thus, the bioavailability and the xenobiotic metabolism are crucial in plant studies as well as in drug discovery and in animal chemical approaches.

However, plant metabolism could have positive applications. Since plants have elaborate chemical mechanisms and strongly select their metabolomes and are polymorphic between accessions, they are particularly well suited for pharmacogenetic variation studies. In particular, across Arabidopsis, accessions have variable sensitivities to chemical compounds; thus, Arabidopsis can be an excellent model to provide insights into the activity of new small molecules [169].

A promising perspective is the possibility of using small molecules with assessed efficacy in plant models for applications on species that are not prone to be manipulated genetically. A recent example is reported by Zhao et al. [109], who addressed the problem of low transformation efficiency in soybean (*Glycine max*), one of the most important crops worldwide. Indeed, in soybean, many agronomic traits show polygenic inheritance; therefore, mutations in different genes are required to obtain a specific phenotype, and this makes the transformation efficiency a key parameter for the genetic improvement of this crop [170]. So far, *Agrobacterium*-mediated transformation is the most effective method in soybean, although the transformation efficiency heavily hinges on the genotypes, depending on their susceptibility to infection [171]. In this work, the authors focused on the impact of plant regeneration in vitro after protoplast transformation and on the methylation-mediated silencing of the transgenes. Following treatment with 5-azaC, the increased methylation level of the transgene, and thus its expression, increased significantly. This was correlated with an increased transformation efficiency of soybean genotypes with both low and high intrinsic transformation efficiency. Moreover, treatment with 5-azaC greatly enhanced the shoot regeneration efficiency, a key step in the transformation process, opening new perspective for the development of improved transformation protocols towards recalcitrant genotypes and species [109].

The enthusiasm for chemical genetics and small molecule tools applied in plants has been fluctuating in the last two decades but constantly increasing. Common efforts in the plant community, such as sharing data and creating tools and database online, e.g., ChemMine tools [172], could improve the overall field. Indeed, it is imperative to establish common principles, as normalised rules and protocols, to provide the field of plant chemical genetics with the same robustness as that in animal studies.

## Data Availability

Not applicable.

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
