# Peer review of "Plants and Small Molecules: An Up-and-Coming Synergy"

_plants, 2023, doi:10.3390/plants12081729_

Round 1
Reviewer 1 Report
The review titled “Plants and small molecules: an up-and-coming synergy” written by Lepri and co-workers focused on the chemical genetics used as a tool for studying plant photomorphogenesis and epigenetic processes. Authors also discussed about plant system as a tool for the identification of small molecules, which can be improved through the chemical synthesis. The data for the last ten years have been reported in a clear and precise manner, furnishing usefull information for the researchers in the field, therefore the manuscript can be accepted for the publication in the journal requiring only minor revisions.
Details:
- - Page 7: specify after DELLA “aspartic acid–glutamic acid–leucine–leucine–alanine” in brackets
- - the compounds shown in the figures must be labelled in the text adding the appropriate figure number in brackets
- - Fig 3: the chemical structure of A1 is not correct, it should be a pyridine-thiazole derivative
- - page 18 and page 20, in the sentences “using in vivo gametophytic development”, “Specific in vivo assays can be designed for target proteins” and “In vivo profiling with these probes”: in vivo should be in italics
Author Response
Point 1: Page 7: specify after DELLA “aspartic acid–glutamic acid–leucine–leucine–alanine” in brackets
Response 1: We are sorry for this oversight. We have added the text in brackets.
Point 2: the compounds shown in the figures must be labelled in the text adding the appropriate figure number in brackets
Response 2: We would like to thank the Reviewer for this suggestion. We have entered the figures number in the text.
Point 3: Fig 3: the chemical structure of A1 is not correct, it should be a pyridine-thiazole derivative
Response 3: We would like to thank the Reviewer for this remark. However, the structure depicted in figure 3 is correct and matches with the one reported in the following reference
“Sukiran, N.A.; Pollastri, S.; Steel, P.G.; Knight, M.R. Plant Growth Promotion by the Interaction of a Novel Synthetic Small Molecule with GA-DELLA Function. Plant direct 2022, 6, e398, doi:10.1002/pld3.398”.
Point 4: page 18 and page 20, in the sentences “using in vivo gametophytic development”, “Specific in vivo assays can be designed for target proteins” and “In vivo profiling with these probes”: in vivo should be in italics
Response 4: We edited the text according to the Reviewer's suggestion

Reviewer 2 Report
Chemical genetics is an alternative to classical genetics and has contributed to numerous scientific discoveries. In the field of plant science, it continues to provide an extremely significant amount of knowledge, particularly regarding the function of plant hormones. Chemical genetics also functions as a powerful tool in the fields of stress tolerance and epigenetics, and has clearly become an indispensable tool in plant research.
The authors carefully review the results of chemical genetics in plant research. In particular, the authors provide sections on photomorphogenesis, abiotic and biotic stresses, and epigenetic regulation. This review is an appropriate summary of the results of chemical genetics in plant research and will be very informative for many readers. The following are comments on this paper
1. In the abstract, it is mentioned that "However, there are still several drawbacks that cannot be easily solved with molecular genetic approaches, such as the study of unfriendly species, which are of increasing agronomic interest, but are not easily transformed, thus are not prone to many molecular techniques. Chemical genetics represents a methodology able to fill this gap. I agree with this statement, but there are very few references to cases other than model plants in the paper. The addition of the last section on the application to difficult-to-transform plants would provide useful information for many plant scientists (who do not use Arabidopsis).
2. Concerning the balance depicted in Figure 1. Carefully describe the meaning of this figure, as it is not clear why classical genetics is depicted as heavier than chemical genetics.
3. The description of auxins in the Introduction is a case in point and should be a separate new section.
Author Response
Point 1: In the abstract, it is mentioned that "However, there are still several drawbacks that cannot be easily solved with molecular genetic approaches, such as the study of unfriendly species, which are of increasing agronomic interest, but are not easily transformed, thus are not prone to many molecular techniques. Chemical genetics represents a methodology able to fill this gap. I agree with this statement, but there are very few references to cases other than model plants in the paper. The addition of the last section on the application to difficult-to-transform plants would provide useful information for many plant scientists (who do not use Arabidopsis).
Response 1: We agree with this convincing suggestion from the Reviewer; accordingly, we have added a paragraph, in the section “Future perspectives and challenges”, citing a work focused on the improvement of soybean transformation by using 5-azaC. In our opinion, this work, among the few in non-Arabidopsis chemical genetics, was the most interesting, from the point of view of unfriendly species.
Point 2: Concerning the balance depicted in Figure 1. Carefully describe the meaning of this figure, as it is not clear why classical genetics is depicted as heavier than chemical genetics.
Response 2: We are sorry for this oversight. We have added the legend to Figure 1, explaining why classical genetics is depicted heavier than chemical genetics.
Point 3: The description of auxins in the Introduction is a case in point and should be a separate new section.
Response 3: We would like to thank the Reviewer for this suggestion. We have split the paragraph on small molecules in auxin signaling from the Introduction, creating a new section, namely "A successful case: small molecules to study auxin”.
